# Changes of Dorsal Root Ganglion Volume in Dogs with Clinical Signs of Degenerative Myelopathy Detected by Water-Excitation Magnetic Resonance Imaging

**DOI:** 10.3390/ani11061702

**Published:** 2021-06-07

**Authors:** Eiji Naito, Kohei Nakata, Yukiko Nakano, Yuta Nozue, Shintaro Kimura, Hiroki Sakai, Osamu Yamato, Md Shafiqul Islam, Sadatoshi Maeda, Hiroaki Kamishina

**Affiliations:** 1Joint Graduate School of Veterinary Sciences, Gifu University, Gifu 501-1193, Japan; nedved3311@gmail.com (E.N.); shiroki@gifu-u.ac.jp (H.S.); sadat@gifu-u.ac.jp (S.M.); 2The Animal Medical Center of Gifu University, Faculty of Applied Biological Sciences, Gifu University, Gifu 501-1193, Japan; k.nakata.1986@gmail.com (K.N.); limetomint@hotmail.com (Y.N.); ytnz.23@gmail.com (Y.N.); shinta_ta_ta@yahoo.co.jp (S.K.); 3The United Graduate School of Veterinary Sciences, Gifu University, Gifu 501-1193, Japan; 4Joint Faculty of Veterinary Medicine Kagoshima University, Kagoshima 890-0065, Japan; osam@vet.kagoshima-u.ac.jp (O.Y.); si.mamun@ymail.com (M.S.I.)

**Keywords:** degenerative myelopathy, dogs, dorsal root ganglion, magnetic resonance imaging, nerve root, water-excitation

## Abstract

**Simple Summary:**

Canine degenerative myelopathy (DM) is a chronic, progressive, and fatal neurodegenerative disease. Although degenerative changes in dogs with DM are observed not only in the spinal cord white matter but also the dorsal root ganglion (DRG) neurons, these changes are undetectable on conventional magnetic resonance imaging (MRI). Therefore, we investigated the ability of water-excitation MRI to visualize the DRG in dogs, and whether volumetry of DRG has a premortem diagnostic value for DM. Using water-excitation MRI, DRG could be depicted in all dogs. To normalize the volumes of DRG, body surface area was the most suitable denominator. The normalized DRG volume in dogs with DM was significantly lower than those in control dogs and dogs with intervertebral disc herniation. The results of this study revealed that widespread atrophy of DRG was likely to occur in DM. Moreover, volume reductions of DRG were observed in dogs with DM in both the early disease stage and late disease stage. Our research suggests that the DRG volume obtained by the water-excitation technique could be used as a clinical biomarker for DM.

**Abstract:**

Canine degenerative myelopathy (DM) is a progressive and fatal neurodegenerative disease. However, a definitive diagnosis of DM can only be achieved by postmortem histopathological examination of the spinal cord. The purpose of this study was to investigate whether the volumetry of DRG using the ability of water-excitation magnetic resonance imaging (MRI) to visualize the DRG in dogs has premortem diagnostic value for DM. Eight dogs with DM, twenty-four dogs with intervertebral disc herniation (IVDH), and eight control dogs were scanned using a 3.0-tesla MRI system, and water-excitation images were obtained to visualize and measure the volume of DRG, normalized by body surface area. The normalized mean DRG volume between each spinal cord segment and mean volume of all DRG between T8 and L2 in the DM group was significantly lower than that in the control and the IVDH groups (*P* = 0.011, *P* = 0.002, respectively). There were no correlations within the normalized mean DRG volume between DM stage 1 and stage 4 (*r_s_* = 0.312, *P* = 0.128, respectively). In conclusion, DRG volumetry by the water-excitation MRI provides a non-invasive and quantitative assessment of neurodegeneration in DRG and may have diagnostic potential for DM.

## 1. Introduction

Canine degenerative myelopathy (DM) is a fatal neurodegenerative spinal cord disorder that develops in several breeds including German Shepherds, Boxers, and Pembroke Welsh Corgis (PWC) [1,2]. The etiology of DM has not yet been fully elucidated; however, a previous study reported that DM-affected dogs were homogeneous for the A allele of a superoxide dismutase 1 (SOD1) missense mutation, SOD1: c.118G > A, which predicts a p.E40K amino acid substitution [3]. Age at the onset of clinical signs was 8 years or older in most cases [4]. The clinical signs of DM initially appear in the pelvic limbs as spastic upper motor neuron paresis and general proprioceptive ataxia, which progresses to flaccid tetraplegia and eventually dyspnea [2,5].

Currently, the clinical diagnosis of DM is based on the following criteria: confirmation of the progression of clinical signs, identification of reported SOD1 mutations, and exclusion of other progressive spinal cord disorders that clinically mimic DM [2,4]. However, a definitive diagnosis of DM can only be achieved by postmortem histopathological examination of the spinal cord [2]. Therefore, novel diagnostic tools with higher specificity to DM are needed in order to make a faster and more accurate premortem diagnosis.

The pathological changes in the spinal cord in DM are characterized by axonal degeneration [3,6,7], axonal loss [3,6,7], demyelination of the white matter in the spinal cord [7,8], loss of thoracic sensory root axons [9,10], and degenerative changes in the dorsal root ganglion (DRG) neurons [10]. Although these marked histopathological changes occur at every part of the spinal cord [7], conventional magnetic resonance imaging (MRI) techniques do not depict DM lesions [7,11]. Currently, technological advances have led to the development of high-field MRI for clear visualization of the spinal cord and the peripheral nerve tissue. The water-excitation technique has been demonstrated to provide better fat suppression and overall better image quality compared with conventional T1-weighted fat saturation [12]. In humans, magnetic resonance neurography with water-excitation was introduced as a modified method for visualizing the peripheral nervous system [13,14]. In addition, a previous study reported that nerve root volume could be measured using three-dimensional fast field echo water-excitation [15].

The first aim of the current study was to establish the normalization of DRG volume in dogs using the water-excitation technique. The associations between DRG volume or spinal cord cross-sectional area and body weight, body surface area, or vertebral body length were evaluated. The second aim was to investigate the diagnostic ability of the water-excitation technique for DM. Dogs with intervertebral disc herniation (IVDH) were included in this study because IVDH is the most common cause of hindlimb paralysis that needs to be differentiated from DM. We hypothesized that the water-excitation technique provides clear depiction of DRG and the volume of DRG is reduced in DM dogs because of widespread degenerative changes and loss of nerve root axons compared to control dogs and dogs with IVDH.

## 2. Materials and Methods

### 2.1. Animals

This study was conducted as a retrospective cross-sectional study. All dogs underwent MRI at the Animal Medical Center of Gifu University between August 2019 and January 2021. All owners signed an informed consent form (approved by the Animal Medical Center of Gifu University and Use Committee, protocol #E20005, #2020-230). First, in order to establish the normalization of DRG volume in dogs, this study included control dogs that had no clinical or imaging evidence of vertebral or spinal cord disorders. Second, this study included dogs with DM, dogs with IVDH, and control dogs in order to compare DRG volume. Control dogs in the first study and the second study were the same population. In the DM group, all dogs were diagnosed with DM according to the following criteria: clinical signs consistent with DM (adult onset, slowly progressive, and non-painful paraparesis progressing to tetraplegia) [2,16], unremarkable findings on spinal cord imaging with conventional MRI sequences (Achieva dStream, Philips, Amsterdam, The Netherlands), and genetic testing confirmed homozygosity for the SOD1 c.118G > A missense mutation (A/A) [17]. All DM-affected dogs had progressive clinical signs for at least a year at the time of manuscript preparation. We also included dogs in the DM group that underwent MRI within 24 h after death, prior to necropsy. The number of dogs that received postmortem MRI is stated in the results section. The owners of the dogs were instructed to store the dogs in a cool condition and place refrigerants over the entire spine in order to minimize postmortem changes until they brought the dogs to us. The disease stage of DM was classified into four clinical stages as previously described [2,7]. The clinical stages were characterized as follows: stage 1, general proprioceptive ataxia and upper motor neuron paraparesis; stage 2, non-ambulatory paraparesis to paraplegia; stage 3, lower motor neuron paraplegia to thoracic limb weakness; and stage 4, lower motor neuron tetraplegia and brainstem signs. All dogs in the IVDH group had thoracolumbar spinal cord compression by herniated intervertebral discs, which were confirmed by MRI with or without subsequent gross confirmation at surgery. Thoracolumbar IVDH cases were graded as previously described [18,19]. The clinical grading of thoracolumbar IVDH was as follows: grade 1, thoracolumbar pain only; grade 2, ambulatory paraparesis; grade 3, non-ambulatory paraparesis; grade 4, paraplegia with positive deep pain sensation; and grade 5, paraplegia with a loss of deep pain sensation. Imaging analyses were performed to rule out the presence of concurrent diseases that may contribute to the neurological status of each dog. Exclusion criteria of this study for the DM or IVDH group were as follows: dogs with an incomplete diagnosis, intracranial disorders, vertebral/spinal cord tumors, and intramedullary or intradural extramedullary lesions that can be detected by conventional MRI.

### 2.2. MRI Sequences

All MRI sequences were acquired using a 3.0-Tesla MRI system with an 8-channel coil as an RF coil and field of view adapted to the size of the animal (Figure A1). For MRI procedures, general anesthesia was induced with intravenous propofol (PROPOFOL injection, Fuji Pharma Co. Ltd., Toyama, Japan) and maintained with a mixture of isoflurane (Isoflurane, Pfizer Inc., New York, NY, USA) in oxygen and room air. In the control group, the protocol consisted of a sagittal and transverse T1-weighted sequence (repetition time (TR)/echo time (TE) 570/13.8 ms; slice thickness 1.5 mm) and T2-weighted sequence (TR/TE 3113/90 ms; slice thickness 1.5 mm). In the DM and IVDH groups, the protocol consisted of a sagittal and transverse T1-weighted sequence (TR/TE 570/13.8 ms; slice thickness 1.5 mm), T2-weighted sequence (TR/TE 3113/90 ms; slice thickness 1.5 mm), and contrast enhanced T1-weighted sequence after intravenous injection of 0.1 mmol/kg of gadodiamide hydrate (OMNISCAN, Daiichi-Sankyo, Tokyo, Japan). Water-excitation imaging parameters were as follows: water-excitation time: 13 msec, repetition time: 10.12–10.24 msec, invention time: 150–170 msec, slice thickness of transverse image: 0.375–0.500 mm, slice thickness of coronal and sagittal images: 1.1–1.2 mm, and sequence flip angle: 30°.

### 2.3. Image Analysis

Image data were analyzed using OsiriX MD version 4.1.2 (OsiriX Pixmeo, Geneva, Switzerland). All images included in the present study were anonymized by H.K. and measurements were performed by E.N. E.N. is a practicing veterinarian who received training in veterinary radiology and neurology for seven years. Using water-excitation images, a stack of sequential image slices that cross-sectioned the DRG were selected for quantification. Water-excitation images included the thoracolumbar spinal cord between the T8 and L2 intervertebral disc levels. Volumetry of DRG was performed as previously described with minor modifications (Figure 1) [20]. Water-excitation images were a continuous image of the nerve roots from the intervertebral foramen to the entry of the spinal cord, including the full volume of the DRG on both sides. The transverse images of the DRG were manually segmented by tracing the borders using software that calculated the cross-sectional area of the DRG based on the number of pixels contained within the traced contour. To overcome the inadequate definition between the DRG and spinal cord profiles at the point where the nerve root began to enter the spinal cord, the boundary between the spinal cord and the DRG was defined by tracing the contour of the spinal cord. DRG volume was calculated by summing the cross-sectional areas (*ai*) of each DRG image, multiplying it by the slice thickness (0.375–0.5 mm) (*ti*), and expressing it in the following formula, according to Cavalieri’s principle:*DRG volume* = ∑*aiti*

Each segment of DRG volume was averaged on the left and right sides. The DRG volume was measured in duplicate and the average value was adopted. To normalize the volumes of DRG, the ratios of body weight, body surface area, and L2 vertebral body length to DRG volume were calculated. Vertebral body length was measured on a sagittal water-excitation image. The cross-sectional area of the spinal cord was measured on a transverse water-excitation image at the center of each vertebral body. To evaluate the diagnostic utility of the DRG volume for DM, normalized DRG were compared with a normalized cross-sectional area of the spinal cord.

### 2.4. Statistical Analyses

Statistical analyses were performed using Easy R software [21]. In the control group, the correlation coefficient (*r_s_*) was calculated by evaluating the correlation between the DRG volumes from T8 through L2 and body weight, body surface area, and vertebral body length from T8 through L2 by Spearman’s rank correlation coefficient. The normalized DRG volumes were compared among the DM, IVDH, and control groups using the Kruskal–Wallis test. Post-hoc comparisons employed the Mann–Whitney U-test with Bonferroni correction. Bilateral differences of DRG volumes were calculated from the absolute value of the left-right DRG ratio. In all analyses, a *P* value of < 0.05 was considered significant.

## 3. Results

### 3.1. Sample Population

The characteristics of all dogs are shown in Table 1. The control group consisted of laboratory animals at Gifu University (*n* = 4) and client-owned dogs (*n* = 4). The client-owned dogs in the control group had transient limb ataxia but no structural lesions in the central nervous system (*n* = 2), intracranial neoplasia (*n* = 1), or idiopathic epilepsy (*n* = 1). Laboratory dogs in the control group had no structural lesions in the central nervous system (*n* = 3) or idiopathic epilepsy (*n* = 1). Breeds in the control group included Beagle (*n* = 4), Boston terrier (*n* = 1), mixed-breed (*n* = 1), French bulldog (*n* = 1), and PWC (*n* = 1). Five dogs were spayed females and three dogs were castrated males. The median body weight, median body surface area, L2 vertebral body length, and median age of the dogs in control group were 11.4 kg (range, 6.6–14.0 kg), 0.49 m^2^ (range, 0.33–0.58 m^2^), 16.9 mm (range, 13.4–18.8 mm), and 6.0 years (range, 2.8–10.0 years), respectively. We included eight DM-affected Pembroke Welsh Corgis in the DM group. The clinical stages of DM were as follows: stage 1 (*n* = 4) and stage 4 (*n* = 4). Four PWCs, which were categorized as stage 4, underwent MRI within 24 h of death (Dog #9, 10, 11, and 12). These dogs were diagnosed with DM based on histopathological examination of the spinal cord. The other four PWCs were diagnosed with DM according to the inclusion criteria. Median body weight, median body surface area, L2 vertebral body length, and median age of the dogs in DM group were 12.3 kg (range, 10.1–16.8 kg), 0.52 m^2^ (range, 0.46–0.63 m^2^), 17.2 mm (range, 16.3–18.0 mm), and 13.2 years (range, 10.8–15.9 years), respectively. There were two spayed females, one intact female, and five castrated males. In the IVDH group, we included 24 dogs diagnosed with thoracolumbar IVDH. The neurological grades were as follows: grade 1 (*n* = 2), grade 2 (*n* = 5), grade 3 (*n* = 4), grade 4 (*n* = 8), and grade 5 (*n* = 5). The locations of IVDHs were T12-T13 (*n* = 12), T13-L1 (*n* = 9), L1-L2 (*n* = 6), T11-T12 (*n* = 4), L2-L3 (*n* = 4), L4-L5 (*n* = 2), and T9-T10, T10-T11, L3-L4, L5-L6, and L6-L7 (*n* = 1). The number of disc herniations in each dog was one (*n* = 15), two (*n* = 4), three (*n* = 1), four (*n* = 3), and five (*n* = 1). Surgical treatment was performed in 15 dogs, and non-surgical treatment was selected in nine dogs. The median body weight, median body surface area, median L2 vertebral body length, and median age of the dogs in the IVDH group were 6.7 kg (range, 3.0–14.1 kg), 0.33 m^2^ (range, 0.20–0.58 m^2^), 14.4 mm (range, 10.0–16.9 mm), and 11.7 years (range, 2.6–15.7 years), respectively. There were three intact females, six spayed females, nine intact males, and six castrated males. Breeds in the IVDH group included: Miniature dachshund (*n* = 11), Toy poodle (*n* = 4), French Bulldog (*n* = 2), Pug (*n* = 2), Border Collie (*n* = 1), Chihuahua (*n* = 1), mixed-breed (*n* = 1), Miniature Schnauzer (*n* = 1), and Pekingese (*n* = 1). The control dogs were significantly younger than the dogs with DM (*P* = 0.009) and IVDH (*P* = 0.012). There was no significant difference in age between the DM and IVDH groups. The dogs with IVDH had significantly lower body weight, body surface area, and L2 vertebral body length than the control dogs (*P* = 0.003, 0.005, 0.009, respectively) and dogs with DM (*P* < 0.001). There were no significant differences in body weight, body surface area, or L2 vertebral body length between the control and DM groups.

### 3.2. Normalization of the DRG Volumes in Control Dogs

There were no significant differences in DRG volume among spinal cord segments. Therefore, we used the mean DRG volume of all spinal cord segments between T8 and L2 in the correlation analyses. The strongest correlation was found between body surface area and mean DRG volume (*r_s_* = 0.792, *P* = 0.024) (Table A1). Body weight was found to have a moderate correlation with DRG volume (*r_s_* = 0.691, *P* = 0.037), but L2 vertebral body length did not have a significant correlation with DRG volume (*r_s_* = 0.612, *P* = 0.176). Therefore, comparisons of DRG volume among the three groups were carried out using body surface area as a denominator for normalization. There were no correlations between the mean cross-sectional spinal cord area of the spinal cord segment and body weight (*r_s_* = −0.048, *P* = 0.911), body surface area (*r_s_* = 0.124, *P* = 0.812), and L2 vertebral body length (*r_s_* = 0.571, *P* = 0.151).

### 3.3. Normalized DRG Volumes between the DM, IVDH, and Control Groups

At each spinal cord segment, normalized DRG volumes were significantly lower in the DM group than in the control group at T9 (*P* = 0.038), T10 (*P* = 0.042), and L2 (*P* = 0.045) (Figure 2 and Table 2). Normalized DRG volumes were also significantly lower in the DM group than the IVDH group at T8 (*P* = 0.009), T9 (*P* = 0.003), T10 (*P* = 0.003), T11 (*P* = 0.010), T12 (*P* = 0.035) T13 (*P* = 0.031), L1 (*P* = 0.041), and L2 (*P* = 0.007) (Figure 2 and Table 2). The normalized mean DRG volume of all spinal cord segments between T8 and L2 was significantly lower in the DM group than in the control group and the IVDH group (*P* = 0.011, *P* = 0.002, respectively; Figure 3 and Table 2).

### 3.4. DRG Volumes in DM Dogs with Different Stages

There was no correlation in the mean normalized DRG volume between DM stage 1 and stage 4 (*r_s_* = 0.312, *P* = 0.128). At each segment, there was also no significant difference in the mean normalized DRG volume between DM stage 1 and stage 4.

### 3.5. Laterality of DRG Size Change

The bilateral difference of mean DRG volume of all spinal cord segments between T8 and L2 was 9.8% (standard deviation [SD] 6.0) in the control group, 15.3% (SD 8.2) in the DM group, and 19.8% (SD 13.2) in the IVDH group, and no significant difference was observed among the three groups (Table A2). In the IVDH group, the bilateral difference of DRG volumes at the lesion (25.3%; SD 14.7) was higher than that at the non-lesion site (15.9%; SD 9.5).

### 3.6. Spinal Cord Cross-Sectional Area

There were no significant differences in mean cross-sectional area of the spinal cord among the three groups. At each spinal cord segment, there was also no significant difference in the cross-sectional area of the spinal cord among the three groups.

## 4. Discussion

The present study demonstrated that the water-excitation images depicted DRG and the nerve roots of dogs. The water-excitation sequence is a fat suppression sequence, which is a selective excitation technique to suppress signals from fat tissues by exploiting the difference between water and fat resonance frequencies. This sequence visualizes DRG clearly due to its high spatial resolution and high signal-to-noise ratio [22]. Slice thickness is thinner with water-excitation than with short tau inversion recovery, and reduced slice thickness improves spatial resolution and better visualization of anatomical details [23]. The water-excitation technique produces thinner slice images and provides a clearer depiction of the DRG and nerve roots [24]. As DRG in dogs is not visualized by conventional MRI sequences, water-excitation MRI has potential to be used as a non-invasive diagnostic test for diseases affecting DRG.

The ratio of DRG volume to body surface area showed a strong positive correlation, which can be used to normalize DRG volume across dogs with different sizes. In a previous study of chronic inflammatory demyelinating polyneuropathy (CIDP) in humans, DRG normalized by body surface area was useful for the diagnosis and assessment of the severity of CIDP [15]. We found that the normalized DRG volume was significantly reduced in the DM group compared with the control and the IVDH group. This suggests that the DRG volume obtained by the water-excitation technique could be used as a clinical biomarker for DM.

In an early study, Wallerian degeneration of the dorsal nerve roots and central chromatolysis of the DRG neurons was reported in dogs with DM [9]. More recently, a decreased number of axons in the T8 dorsal nerve root and degenerative changes of DRG neurons in DM-affected dogs have been reported [10]. In mouse models of amyotrophic lateral sclerosis and diabetic neuropathy, axonal degeneration caused impaired axonal transport of proteins and metabolites, resulting in neuronal cell death [25,26]. Therefore, it is considered that nerve cell death due to axonal degeneration occurs in the DRG of DM-affected dogs, leading to decreased DRG volume. Although the histopathological findings of T9-L2 DRG in DM-affected dogs remain unknown, development of concurrent degeneration of DRG neurons in other regions is more likely, given the widespread axonal loss and demyelination in the white matter not only in the caudal thoracic spinal cord but also in the cervical and lumbar spinal cords in DM dogs [3,6,7]. In particular, since degenerative lesions are located in the dorsal and lateral funiculus of the spinal cord through which the axons of DRG neurons pass, widespread atrophy of DRG is likely to occur in DM. 

There was no significant difference in DRG volume between the early and late disease stages in this study. This finding was in contrast with the observation that the C7 dorsal roots of dogs with DM gradually decreased in number with disease progression [27]. A previous study showed that hyporeflexia of the patellar reflex was described in dogs with DM at early disease stage, which was accounted for by the degenerative change of dorsal nerve root and central chromatolysis of DRG neurons [9]. Therefore, DRG volumes may decrease even at the early disease stage. This finding favors an early diagnosis of DM with this non-invasive MRI technique. On the other hand, a decrease in DRG volume may not correlate with lesion load of the spinal cord and therefore may not be suitable for longitudinal monitoring of pathological progression.

The initial clinical signs of DM share similarities with other progressive spinal cord disorders. IVDH is the most common spinal cord disorder in dogs that are also predisposed to DM; therefore, we included dogs with IVDH as a “disease control” in this study. Our study revealed that DRG volume measurement using water-excitation MRI was capable of distinguishing DM from IVDH. Normalized DRG volumes in all spinal cord segments and mean DRG volume were significantly lower in the DM group than in the IVDH group. The reduction of DRG volumes in DM occurred in multiple spinal segments that parallel the diffuse degeneration of the spinal cord, whereas the DRG volume changes were focal in the IVDH group, decreasing in the proximity of the lesion site. In human and rodent studies, ipsilateral DRG at the site of spinal cord injury was atrophied due to demyelination and Wallerian degeneration of the axons of DRG, resulting in dying back degeneration and death of sensory neurons [20,28]. The duration and severity of the disease also had an impact on DRG size as atrophied DRG recovered its size over time after injury [29,30,31]. The wide range of DRG sizes in the IVDH group may be attributed to disease duration, location, and severity of injury in this study.

Several limitations of the present study should be considered. Measurements of DRG were performed and analyzed by a single observer. Although measurements were performed in duplicate, further study is needed to evaluate intra- and inter-observer errors. The sample size was small, especially the number of dogs in the control and the DM groups. In the DM group, MRI data for four dogs were obtained postmortem in order to compare the difference of DRG volumes between the early stage and late stage. This comparison was only possible by using postmortem MRI as dogs with DM in the late stage suffers from respiratory disfunction that hinders diagnostics requiring general anesthesia. Postmortem MRI was performed within 24 h of death in an attempt to minimize any postmortem changes; however, postmortem changes in DRG must be investigated in terms of their effects on histopathological changes and MRI data. In the central nervous system, the previous study showed that comparing premortem and postmortem MRIs for cerebral microbleeds yielded comparable imaging performance [32]. We stored these dogs in a cool condition immediately after death in order to minimize potential postmortem changes. We considered that the obtained MRI data were as close to the premortem state as possible. The other four dogs were tentatively diagnosed with DM without histopathological confirmation. These four dogs were still alive at the time of manuscript preparation; therefore, this study could not compare the histopathological findings of DM with MRI. All dogs in the DM group were PWCs in the present study. In Japan, DM is most common in Corgis, and the number of other breeds that are prone to develop DM is small. The fact that dogs in the DM group only included a single breed was one of the limitations of this study. Dogs in the control group were significantly younger than those of the other two groups. Although a previous study reported no difference in the volume of DRG with age in humans [33], the relationship between DRG volume and age in dogs warrants further investigation. 

## 5. Conclusions

Water-excitation was a useful technique for DRG volumetric analysis in dogs with DM. Volumetry of normalized DRG by the water-excitation technique provided a non-invasive and quantitative assessment of neurodegeneration in DRG and may have diagnostic potential for DM.

## Figures and Tables

**Figure 1 animals-11-01702-f001:**
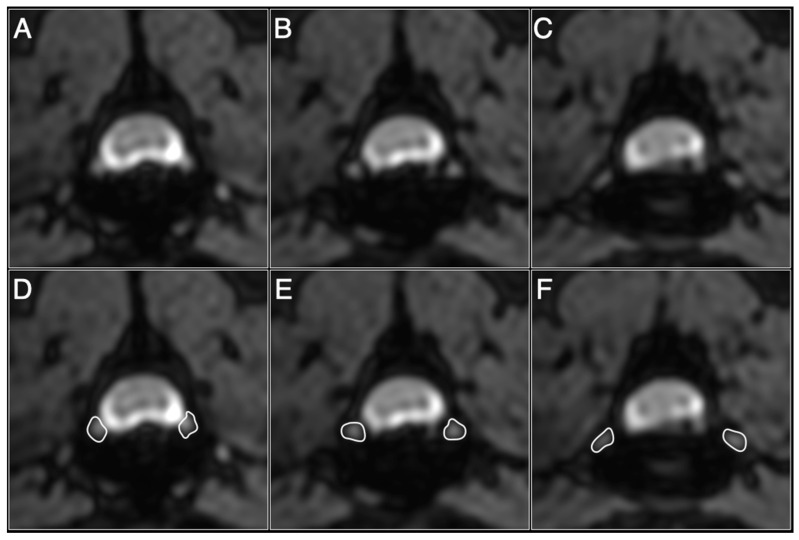
Sequential image slices of dorsal root ganglion (DRG) in control dog (Dog#8). Nerve roots emerge from the spinal cord (**A**), form a DRG that passes caudally (**B**), and enter the intervertebral foramen (**C**). All DRG were manually segmented by tracing the border (white lines) and the cross-sectional area of the DRG was calculated from the number of pixels contained within the traced contour (**D**–**F**).

**Figure 2 animals-11-01702-f002:**
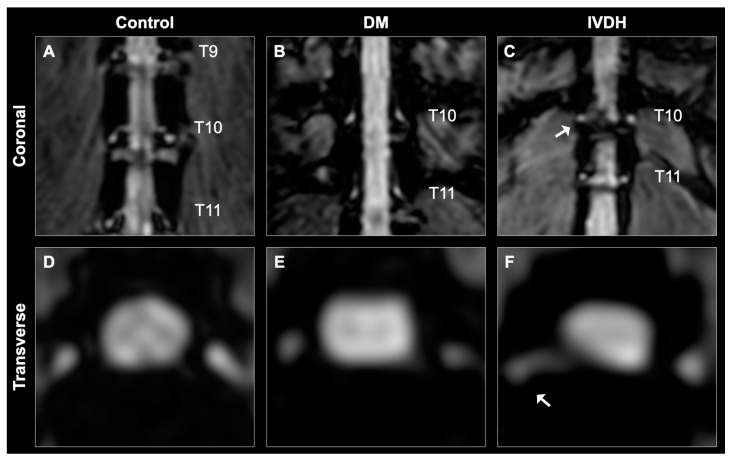
Dorsal root ganglion (DRG) and nerve root from a coronal and transverse magnetic resonance imaging data set. Volumes of DRG were measured from the images acquired by water-excitation sequence. Coronal images (**A**–**C**) showed the caudal thoracic spinal cord and nerve roots, and transverse images (**D**–**F**) showed T10 DRG. The volume of DRG was normal for a control dog (Dog #8) (**A**,**D**), and atrophied in a dog with degenerative myelopathy (Dog #12) (**B**,**E**). Volumetric atrophy of DRG (white arrow) was observed at the herniation site in a dog with inter vertebral disc herniation (Dog #38) (**C**,**F**).

**Figure 3 animals-11-01702-f003:**
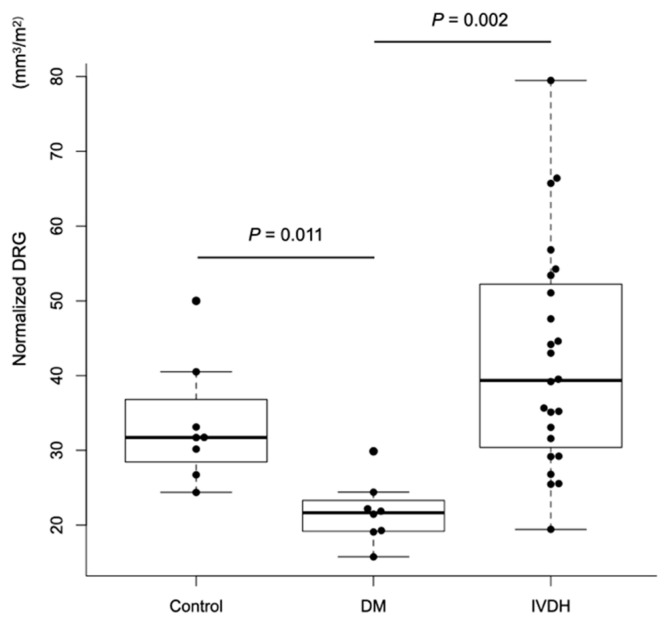
Comparison of the normalized mean dorsal root ganglion (DRG) volume among control, degenerative myelopathy (DM), and intervertebral disc herniation (IVDH) groups. The normalized mean DRG volume in DM group was lower than those in control and IVDH groups (*P* = 0.011 and 0.002, respectively). Horizontal bars indicate medians within groups.

**Table 1 animals-11-01702-t001:** Characteristics of clinical samples ^1^.

Dog Number	Group	Breed	Age (year)	Body Weight (kg)	Body Surface Area (m^2^)	L2 Vertebral Body Length (mm)	Gender	Clinical Duration
1	Control	Beagle	5.0	16.8	0.63	18.8	CM	-
2	Control	Beagle	5.0	13.5	0.55	17.5	SF	-
3	Control	Beagle	5.1	14.0	0.58	16.8	CM	-
4	Control	Beagle	5.5	12.3	0.52	17.3	SF	-
5	Control	Cross-bred	2.8	10.6	0.46	14.9	CM	-
6	Control	FB	8.0	11.4	0.49	12.3	SF	-
7	Control	BT	6.6	6.6	0.33	10.0	SF	-
8	Control	PWC	10.7	10.7	0.46	17.0	SF	-
9	DM-confirmed	PWC	13.0	13.1	0.54	16.9	CM	3 years
10	DM-confirmed	PWC	15.2	10.2	0.46	17.0	CM	3.5 years
11	DM-confirmed	PWC	14.1	12.3	0.52	17.5	CM	3 years
12	DM-confirmed	PWC	15.9	14.6	0.59	18.0	IF	3.3 years
13	DM-suspected	PWC	14.1	11.8	0.49	17.6	CM	2 months
14	DM-suspected	PWC	12.7	13.6	0.55	17.5	CM	3 months
15	DM-suspected	PWC	11.0	12.3	0.52	16.3	SF	7 months
16	DM-suspected	PWC	10.8	16.8	0.63	16.9	SF	3 months
17	IVDH	Pag	11.7	6.1	0.33	12.2	CM	3 months
18	IVDH	MD	5.6	3.0	0.20	14.0	SF	1.5 months
19	IVDH	MD	9.8	4.6	0.25	15.4	IM	1 month
20	IVDH	TP	5.0	6.7	0.33	14.8	IM	11 days
21	IVDH	TP	15.7	3.1	0.20	11.8	IM	5 months
22	IVDH	MS	12.6	9.6	0.43	14.8	IM	2 months
23	IVDH	MD	15.4	7.2	0.36	15.4	SF	6 days
24	IVDH	MD	7.0	9.7	0.43	16.4	IM	4 days
25	IVDH	MD	3.6	5.2	0.29	12.8	SF	3 days
26	IVDH	Cross-bred	11.7	3.4	0.20	10.0	CM	5 days
27	IVDH	MD	13.4	6.5	0.33	15.5	CM	11 days
28	IVDH	FB	9.8	11.6	0.49	16.5	IF	2 months
29	IVDH	TP	12.3	3.4	0.20	14.6	IM	1 month
30	IVDH	MD	5.0	3.4	0.20	12.5	IF	7 days
31	IVDH	TP	5.9	4.2	0.25	13.5	IM	15 days
32	IVDH	MD	4.8	4.5	0.25	12.9	IF	2 days
33	IVDH	FB	7.2	12.8	0.52	16.6	CM	1 month
34	IVDH	Pag	9.7	7.7	0.36	12.7	SF	1 year
35	IVDH	MD	10.1	7.1	0.36	16.0	CM	21 days
36	IVDH	Chihuahua	6.1	5.1	0.29	14.9	IM	7 days
37	IVDH	MD	10.4	9.5	0.43	16.7	CM	2 days
38	IVDH	Pekingese	2.6	5.5	0.29	11.3	IM	9 days
39	IVDH	MD	10.9	5.8	0.29	16.8	SF	1 month
40	IVDH	BC	14.1	14.1	0.58	16.9	SF	1 year

^1^ Abbreviations: DM, Degenerative myelopathy; IVDH, intervertebral disc herniation; FB, French bulldog; BT, Boston terrier; PWC, Pembroke Welsh Corgi; MD, Miniature Dachshund; TP, Toy Poodle; MS, Miniature Schnauzer; BC, Border Collie CM, Castrated male; IM, Intact male; SF, Spayed female; IF, Intact female.

**Table 2 animals-11-01702-t002:** Normalized DRG volumes among control, DM, and IVDH groups.

	Location	Control		DM		IVDH		*P* Value			
		Median	Range	Median	Range	Median	Range	3 Groups	Control vs. DM	Control vs. IVDH	DM vs. IVDH
**Normalized DRG Volume (mm^3^/mm^2^)**	T8	31.841	23.744–51.879	19.079	15.891–35.434	42.604	20.225–72.484	* 0.012	0.155	1	* 0.009
	T9	28.783	23.116–51.924	20.658	15.042–28.717	41.008	21.178–79.567	* 0.004	* 0.038	0.510	* 0.003
	T10	30.808	25.073–50.155	20.943	15.008–26.586	38.804	21.391–81.383	* 0.003	* 0.042	0.858	* 0.003
	T11	31.608	24.592–50.220	21.191	13.516–29.705	38.841	21.257–78.559	* 0.011	0.071	1	* 0.010
	T12	33.454	23.287–53.799	19.653	15.170–31.544	37.888	14.213–86.772	* 0.028	0.071	1	* 0.035
	T13	31.539	24.292–49.628	19.297	14.676–26.998	40.683	12.657–84.264	* 0.028	0.093	1	* 0.031
	L1	30.557	25.384–46.635	20.339	15.889–37.391	35.964	19.987–87.816	* 0.032	0.155	1	* 0.041
	L2	33.220	24.800–45.706	20.205	16.681–34.739	41.609	22.413–72.602	* 0.008	* 0.045	1	* 0.007
	T8-L2	31.727	24.384–49.993	20.373	15.763–29.882	39.360	19.436–79.480	* 0.002	* 0.011	0.510	* 0.002

Abbreviations: DRG, dorsal root ganglion; DM degenerative myelopathy; IVDH, intervertebral disc herniation. * *P* < 0.05.

## Data Availability

Data are available upon request from the authors.

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
