# Peer review of "Changes of Dorsal Root Ganglion Volume in Dogs with Clinical Signs of Degenerative Myelopathy Detected by Water-Excitation Magnetic Resonance Imaging"

_animals, 2021, doi:10.3390/ani11061702_

Round 1

Reviewer 1 Report

The title refers to the use of water-excitation magnetic resonance imaging in dogs diagnosed with MD, but 4 of 8 patients in the study have only a clinical diagnosis of MD. It may be helpful to change the title "Changes of dorsal root ganglion volume to dogs with degenerative myelopathy detected by water-excitation magnetic resonance imaging" in Changes of dorsal root ganglion volume in dogs with clinical signs of degenerative myelopathy detected by water-excitation magnetic resonance imaging"

Line 98 could be rephrased 

Four out eight dog of the DM group underwent MRI within 24 hours after death, prior to necrops

Line 138 – 139 -  It is not clear if the area has been traced only by one operator as in this case the method need to be discussed as the dimension of the dedected area is the focal point of the whole study. There are several reports on reducing measurement errors by using multiple operators. This limitation need to be added in the discussion at line 314 before the considerations about the sample size.

Author Response

Manuscript ID: animals-1226764

Dear Prof. Reviewer 1,

We appreciate the reviewers’ feedback and have addressed the reviewers’ comments. Our responses to the comments are listed below along with the locations of the changes made in the revised manuscript.

I have attached the revised manuscript entitled: “Changes of dorsal root ganglion volume in dogs with degenerative myelopathy detected by water-excitation magnetic resonance imaging”. Our revisions are highlighted in the manuscript.

Please contact me should you have any questions or suggestions regarding the revised manuscript.

We look forward to hearing from you.

Sincerely,

Eiji Naito

Joint Graduate School of Veterinary Sciences, Gifu University, Gifu, Japan

Comments from the editors and reviewers:

-Reviewer 1 Comment No. 1

The title refers to the use of water-excitation magnetic resonance imaging in dogs diagnosed with MD, but 4 of 8 patients in the study have only a clinical diagnosis of MD. It may be helpful to change the title "Changes of dorsal root ganglion volume to dogs with degenerative myelopathy detected by water-excitation magnetic resonance imaging" in Changes of dorsal root ganglion volume in dogs with clinical signs of degenerative myelopathy detected by water-excitation magnetic resonance imaging"

Our response:

We agree with you and have incorporated this suggestion into the revised title.

This change has been made in the following location:

Title, line 2-3

-Reviewer 1 Comment No. 2

Line 98 could be rephrased 

Four out eight dogs of the DM group underwent MRI within 24 hours after death, prior to necropsy.

Our response:

We made changes to this sentence as follows: We also included dogs in the DM group that underwent MRI within 24 hours after death, prior to necropsy.

The number of dogs that received postmortem MRI was stated in the result section (Results, line 194).

This change has been made in the following location:

Materials and Methods, line 101-102

-Reviewer 1 Comment No. 3

Line 138 – 139 - It is not clear if the area has been traced only by one operator as in this case the method needs to be discussed as the dimension of the detected area is the focal point of the whole study. There are several reports on reducing measurement errors by using multiple operators. This limitation needs to be added in the discussion at line 314 before the considerations about the sample size.

Our response:

We agree with reviewer 1 and have incorporated this suggestion into the revised manuscript. In the present study, all images were anonymized by H.K. and measurements were performed by E.N. In order to reduce the intra-observer error, measurements were performed in duplicate and the changes were made in the revised manuscript. We did not perform measurements by multiple observers and this was one of the limitations of this study. We have added above information, new results, and the limitation about the inter-observer error.

These changes have been made in the following locations:

These changes have been made in the following locations:

Abstract, line 37-39

Materials and Methods, line 139-140, line 156-157

Results, line 228-230, 234-235, 238-241, 243-244, 258-259, 262, 266-270

Discussion, line 334-337

Table 2

Figure 3

Table A1

Table A2

Reviewer 2 Report

Very Interesting and original study.

I would like to highlight some mistakes that need correction and suggestions which could help improving the quality of the submitted manuscript.

Graphical abstract:

-Authors are encouraged to provide a graphical abstract to improve understanding and quality of the submitted manuscript.

Keywords:

-Line 42: The keyword "body surface area" does not represent the study. Authors are advised to include “magnetic resonance imaging” as a keyword.

Introduction:

-Line 78-80: This information should only be included in the materials and methods section, it is not part of the introduction.

 -Line 81-84: You are not showing a hypothesis, you are stating the obtained results from  your study.

-No information about IVDH is included in this section when authors have decided to include a group of animals with IVDH.

Materials and methods:

-lines 99-100: Why were dogs in the DM group that underwent MRI within 24 hour after death included in the study?

Results:

-Line 176: Please write “central nervous system”.

-Line 185-186: Could the fact that authors evaluated dead animals affect the obtained results?

-Why were different numbers of dogs included in each group instead of including same number? This fact does not allow to compare obtained results in between groups in an equal way.

Discussion:

-Little discussion is provided, authors are encouraged to discuss their results in more detail and compare them with more studies to improve the quality of their manuscript.

-Line 315-319: As described by the authors a big limitation of this study is the fact that 50% of data from DM group were obtained post-mortem with no histopathological study to evaluate differences.

Author Response

Dear Prof. Reviewer 2,

We appreciate the reviewers’ feedback and have addressed the reviewers’ comments. Our responses to the comments are listed below along with the locations of the changes made in the revised manuscript.

I have attached the revised manuscript entitled: “Changes of dorsal root ganglion volume in dogs with degenerative myelopathy detected by water-excitation magnetic resonance imaging”. Our revisions are highlighted in the manuscript.

Please contact me should you have any questions or suggestions regarding the revised manuscript.

We look forward to hearing from you.

Sincerely,

Eiji Naito

Joint Graduate School of Veterinary Sciences, Gifu University, Gifu, Japan

Comments from the editors and reviewers:

-Reviewer 2 Comment No. 1

Authors are encouraged to provide a graphical abstract to improve understanding and quality of the submitted manuscript.

Our response:

Thank you for your suggestion. We have added a graphical abstract to the appendix.

-Reviewer 2 Comment No. 2

Line 42: The keyword "body surface area" does not represent the study. Authors are advised to include “magnetic resonance imaging” as a keyword.

Our response:

We deleted "body surface area" and added “magnetic resonance imaging” to Keywords.

This change has been made in the following location:

Keywords, line 42

-Reviewer 2 Comment No. 3

Line 78-80: This information should only be included in the materials and methods section, it is not part of the introduction.

Our response:

Thank you for your comment. We deleted this sentence from introduction.

-Reviewer 2 Comment No. 4

Line 81-84: You are not showing a hypothesis, you are stating the obtained results from your study.

Our response:

Thank you for your suggestion. We changed our hypothesis as follows: We hypothesized that the water-excitation technique provides clear depiction of DRG and the volume of DRG is reduced in DM dogs because of widespread degenerative changes and loss of nerve root axons compared to control dogs and dogs with IVDH.

This change has been made in the following location:

Introduction, line 80-83

-Reviewer 2 Comment No. 5

No information about IVDH is included in this section when authors have decided to include a group of animals with IVDH.

Our response:

Thank you for your comment. We agree with you and have incorporated this suggestion into the revised manuscript. IVDH cases were included as a “disease control” because IVDH is a common cause of hindlimb paralysis which needs to be differentiated from DM.

We added the following sentence:

Dogs with intervertebral disc herniation (IVDH) was included in this study because IVDH is the most common cause of hindlimb paralysis that needs to be differentiated from DM.

This change has been made in the following location:

Introduction, line 78-80

-Reviewer 2 Comment No. 6

lines 99-100: Why were dogs in the DM group that underwent MRI within 24 hour after death included in the study?

Our response:

Postmortem MRI was performed in 4 dogs with DM. The data of these dogs provided DRG volumes of DM in the late stage and made it possible to compare the difference of DRG volumes between the early stage and late stage. This comparison is only possible by using postmortem MRI as dogs with DM in the late stage suffers from respiratory disfunction that hinders diagnostics requiring general anesthesia. We used dogs within 24 hour after death to minimize postmortem changes, however the effect of such changes on DRG volumes were not investigated in the present study. Another advantage of using the postmortem MRI was that an autopsy could be performed immediately after MRI and definitive diagnosis be made by histopathological examination. We added the rationale of the use of postmortem MRI and its limitation.

These changes appear in:

Discussion, line 338-342, 344-349

-Reviewer 2 Comment No. 7

Line 176: Please write “central nervous system”.

Our response:

We have carefully revised the manuscript and checked for accuracy.

Changes have been made in the following locations:

Results, line 186 and 187

-Reviewer 2 Comment No. 8

-Line 185-186: Could the fact that authors evaluated dead animals affect the obtained results?

Our response:

The owners of the dogs were instructed to store the dogs in a cool condition and place refrigerants over the entire spine in order to minimize post-mortem changes until they bring the dogs to us. As mentioned in our response to Reviewer 2 Comment No. 6, the effect of postmortem changes on DRG volumes were not investigated in the present study and this was one of the limitations of this study.

This change has been made in the following location:

Materials and Methods, line 102-104

Discussion, line 338-342, 344-349

-Reviewer 2 Comment No. 9

Why were different numbers of dogs included in each group instead of including same number? This fact does not allow to compare obtained results in between groups in an equal way.

Our response:

Thank you for your suggestion. This study was conducted in a retrospective manner. Since we collected data of dogs that met our inclusion criteria during the study period, the numbers of dogs were different in three groups. We have included a new sentence regarding this point. We performed statistical analyses with more stringent criteria (i.e. a P value set at < 0.03 as a statistical significance), and the results still support our conclusion that the reduction of DRG volumes in DM occurred in multiple spinal segments and reduced DRG volume can be a biomarker of DM.

These changes have been made in the following locations:

Materials and Methods, line 86-88

-Reviewer 2 Comment No. 10

Little discussion is provided, authors are encouraged to discuss their results in more detail and compare them with more studies to improve the quality of their manuscript.

Our response:

We revised the manuscript according to your recommendations.

First, we have added a description about the utility of the water-excitation sequence in the visualization of DRG. The water-excitation technique produces thinner slice images and provides a clearer depiction of the DRG and nerve roots.

Second, a previous study of chronic inflammatory demyelinating polyneuropathy (CIDP) in humans showed that the ratio of DRG to body surface area detected by MRI was useful for the diagnosis and assessment of the severity of CIDP.

These changes have been made in the following locations:

Discussion, line 277-284, line 288-291

-Reviewer 2 Comment No. 11

Line 315-319: As described by the authors a big limitation of this study is the fact that 50% of data from DM group were obtained post-mortem with no histopathological study to evaluate differences.

Our response:

As stated in our response to Reviewer 2 Comment No. 6, postmortem MRI was performed in 4 dogs with DM. Inclusion of post-mortem MRI was to compare the DRG volumes between dogs with DM at the early and late stages. This comparison was clinically important to know whether the DRG volumes become reduced or unchanged over time as disease progresses. This comparison was only possible by using postmortem MRI as dogs with DM in the late stage suffers from respiratory disfunction that hinders diagnostics requiring general anesthesia. The owners were instructed to minimize post-mortem changes as much as possible and we used the dogs within 24 hour after death. Another advantage of using the postmortem MRI was that an autopsy could be performed immediately after MRI and definitive diagnosis be made by histopathological examination. However, the potential effects of post-mortem changes on DRG volumes cannot be ruled out and this was one of the limitations of our study. We added the rationale of the use of postmortem MRI and its limitation.

These changes have been made in the following locations:

Discussion, line 338-342, 350-352

Reviewer 3 Report

Dear authors,

thank you for submitting this interesting article on the use of water excitation MRI to detect changes in the dorsal ganglion in dogs with degenerative myelopathy. 
These are my comments:

The abstract is concise, but clear and explanatory.
The Introduction gives a fair description of the background research in the subject of DM. I think it would be relevant to specify slightly more in detail why the water excitation techniques provide better fat suppression and overall  image quality. This can be done either in the introduction or in the discussion.
The aims are clearly explained.

Matherials and methods.
 - Either here or in the introduction you should specify the type of study: retrospective or prospective; It is not clear if you looked at previous cases or recruited all animals presented with neurological signs consistent with DM or IVDH. 
- What is the time range for when you collected your cases in the different groups? (if it was retrospective) 
- Where there exclusion criteria? This needs to be specified.
- You mention a group of control dogs twice; were there one or two control groups? (Lines 89 to 92)

- On Image analysis section : how many people performed the measurements? Were they blinded to the type of pathology/group of dogs?How many times was each measurement performed? Was there any statistics done on the inter- or intra-observer repeatability? If the person performing the measurements was not blinded there might be a large bias and this needs to be clarified or corrected by repeating the measurements.

- All your DM cases were Corgis. Was this done on purpose or is it a reflection of your patient load? this must be specified, as it might bias the results of the measurements (since all Corgis are similar in body size).
- I also believe it would be important to provide at least one picture comparing the visibility of the dorsal ganglion with the water excitation and with the standard sequences. Since this is the base of the whole project.

Results
- Line 185 How where the dead dogs kept for 24 hours until the MR was performed? Were they frozen? This might have affected the size of the ganglions.

- Line 192 : when were the MR performed in the IVDH group comparing to the presentation of clinical signs? How many of these dogs had the herniation confirmed surgically?

Discussion:

-Line 300: you need to give a reference for such a strong statement as well as a possible explanation for this fact (if there is one in the literature).
- Amongst the limitations you have to mention the fact that all DM dogs were Corgis.
- Line 3176: this section should be expanded; Of your 8 dogs with DM, you only had representatives for stage 1 and stage 4 and all the stage 4 were post mortem MRI. How can yo be sure that there were no post mortem changes to the size of the ganglion? You should provide at least one reference about post mortem changes affecting the appearance of spinal cord and the ganglions in MRI. If there is (as one would expect) reduction of the spinal cord after death, your sample of dogs would be only the 4 alive cases, so it is essential to justify using the post mortem cases.

Author Response

Dear Prof. Reviewer 3,

We appreciate the reviewers’ feedback and have addressed the reviewers’ comments. Our responses to the comments are listed below along with the locations of the changes made in the revised manuscript.

I have attached the revised manuscript entitled: “Changes of dorsal root ganglion volume in dogs with degenerative myelopathy detected by water-excitation magnetic resonance imaging”. Our revisions are highlighted in the manuscript.

Please contact me should you have any questions or suggestions regarding the revised manuscript.

We look forward to hearing from you.

Sincerely,

Eiji Naito

Joint Graduate School of Veterinary Sciences, Gifu University, Gifu, Japan

Comments from the editors and reviewers:

-Reviewer 3 Comment No. 1

The abstract is concise, but clear and explanatory.

The Introduction gives a fair description of the background research in the subject of DM. I think it would be relevant to specify slightly more in detail why the water excitation techniques provide better fat suppression and overall image quality. This can be done either in the introduction or in the discussion.

The aims are clearly explained.

Our response:

We have added the sentence regarding the utility of water-excitation sequence in the visualization of DRG. The water-excitation sequence is one of fat suppression sequences, which is a selective excitation technique to suppress signals from fat tissues by exploiting the difference between water and fat resonance frequencies. This sequence visualizes DRG clearly due to its high spatial resolution and high signal-to-noise ratio [22]. Slice thickness is thinner with water-excitation than with short tau inversion recovery, and reduced slice thickness improves spatial resolution and better visualization of anatomical details. The water-excitation technique produces thinner slice images and provides a clearer depiction of the DRG and nerve roots.

These changes have been made in the following manuscript locations:

Discussion, line 277-284

-Reviewer 3 Comment No. 2

Either here or in the introduction you should specify the type of study: retrospective or prospective; It is not clear if you looked at previous cases or recruited all animals presented with neurological signs consistent with DM or IVDH. 

Our response:

Thank you for your suggestion. In the revised manuscript, we stated that this study was a retrospective cross-sectional study.

This change has been made in the following location:

Materials and Methods, line 86

-Reviewer 3 Comment No. 3

What is the time range for when you collected your cases in the different groups? (if it was retrospective) 

Our response:

This study was conducted as a retrospective cross-sectional study, using animals administered to our hospital for MRI between August 2019 and January 2021.

This change has been made in the following location:

Materials and Methods, line 87-88

-Reviewer 3 Comment No. 4

Where there exclusion criteria? This needs to be specified.

Our response:

Thank you for your suggestion. We added a sentence specifying our exclusion criteria as follows: Exclusion criteria of this study for the DM or IVDH group were as follows: dogs with an incomplete diagnosis, intracranial disorders, vertebral/spinal cord tumors, and intramedullary or intradural extramedullary lesions which can be detected by conventional MRI.

This change has been made in the following location:

Materials and Methods, line 116-119

-Reviewer 3 Comment No. 5

You mention a group of control dogs twice; were there one or two control groups? (Lines 89 to 92)

Our response:

Thank you for your suggestion. Control dogs in the first study and the second study were the same population.

This change has been made in the following location:

Materials and Methods, line 93-94

-Reviewer 3 Comment No. 6

On Image analysis section : how many people performed the measurements? Were they blinded to the type of pathology/group of dogs? How many times was each measurement performed? Was there any statistics done on the inter- or intra-observer repeatability? If the person performing the measurements was not blinded there might be a large bias and this needs to be clarified or corrected by repeating the measurements.

Our response:

We agree with reviewer 3 and have incorporated this suggestion into the revised manuscript. In the original manuscript, all images were anonymized by H.K. and measurements were performed by E.N. (for some reason, this particular sentence was deleted in the submitted manuscript). E.N. repeated DRG volume measurements in order to reduce the intra-observer error, and we reflected new results in the revised manuscript. Measurements were not performed by multiple observers and this was one of the limitations of this study. We have added above information, results, and the limitation about the inter-observer error.

These changes have been made in the following locations:

Abstract, line 37-39

Materials and Methods, line 139-140, line 156-157

Results, line 228-230, 234-235, 238-241, 243-244, 258-259, 262, 266-270

Discussion, line 334-337

Table 2

Figure 3

Table A1

Table A2

-Reviewer 3 Comment No. 7

All your DM cases were Corgis. Was this done on purpose or is it a reflection of your patient load? this must be specified, as it might bias the results of the measurements (since all Corgis are similar in body size).

Our response:

DM is most common in Corgis in Japan and the number of the case load of other breeds that are prone to develop DM is small. As pointed out by the reviewer, the fact that dogs in DM group only included a single breed was one of the limitations of this study and therefore we added a following sentence to the limitation statement in discussion.

This change has been made in the following locations:

Discussion, line 352-355

-Reviewer 3 Comment No. 8

I also believe it would be important to provide at least one picture comparing the visibility of the dorsal ganglion with the water excitation and with the standard sequences. Since this is the base of the whole project.

Our response:

Thank you for your suggestion. We added an appendix figure showing the difference in visibility of DRG with standard MRI sequences and water-excitation MRI.

This change has been made in the following locations:

Materials and Methods, line 122

Figure A1

-Reviewer 3 Comment No. 9

Line 185 How where the dead dogs kept for 24 hours until the MR was performed? Were they frozen? This might have affected the size of the ganglions.

Our response:

The owners of the dogs were instructed to store the dogs in a cool condition and place refrigerants over the entire spine in order to minimize post-mortem changes until they bring the dogs to us, but not frozen, until MRI was performed. The effect of postmortem changes on DRG volumes were not investigated in the present study and this was one of the limitations of this study. We added a description of how the dogs were kept until the MRI was performed as follows: The owners of the dogs were instructed to store the dogs in a cool condition and place refrigerants over the entire spine in order to minimize postmortem changes until they bring the dogs to us.

This change has been made in the following location:

Materials and Methods, line 102-104

Discussion, line 346-349

-Reviewer 3 Comment No. 10

Line 192 : when were the MR performed in the IVDH group comparing to the presentation of clinical signs? How many of these dogs had the herniation confirmed surgically?

Our response:

We have added the clinical duration for dogs in DM and IVDH group in the Table 1. In the IVDH group, surgical treatment was performed in 15 dogs and non-surgical treatment was selected in 9 dogs.

These changes have been made in the following locations:

Table 1

Results, line 207-208

-Reviewer 3 Comment No. 11

Line 300: you need to give a reference for such a strong statement as well as a possible explanation for this fact (if there is one in the literature).

Our response:

Since the initial symptom of DM is hindlimb paralysis, we selected IVDH, which is the most common disease causing hindlimb paralysis, was used as the disease control group.

This sentence appears in the following locations:

Introduction, line 78-80

-Reviewer 3 Comment No. 12

Amongst the limitations you have to mention the fact that all DM dogs were Corgis.

Our response:

As mentioned in our response to Reviewer 3 Comment No. 7, we added a sentence in limitation section in discussion that all DM dogs were Corgis.

This sentence appears in the following locations:

Discussion, line 352-355

-Reviewer 3 Comment No. 13

Line 3176: this section should be expanded; Of your 8 dogs with DM, you only had representatives for stage 1 and stage 4 and all the stage 4 were post mortem MRI. How can yo be sure that there were no post mortem changes to the size of the ganglion? You should provide at least one reference about post mortem changes affecting the appearance of spinal cord and the ganglions in MRI. If there is (as one would expect) reduction of the spinal cord after death, your sample of dogs would be only the 4 alive cases, so it is essential to justify using the post mortem cases.

Our response:

Thank you for your suggestion. There are no reports on postmortem changes of DRG on MR images. A previous study showed that comparing premortem and postmortem MRIs for cerebral microbleeds yielded comparable imaging performance (Haller S, PLoS One, 2016).

.

This change has been made in the following location:

Discussion, line 344-349

Other changes were described in the manuscript as follows:

Materials and Methods, line 89

Discussion, line 296, 308, 326, 328, 352

References, line 387-471

Round 2

Reviewer 3 Report

Dear authors

thank you for addressing all my comments. I am satisfied with your consequent  editing of the paper.

One last comment: what is the role of the person performing the measurements ( E.N. line 140)? (Neurologist, radiologist etc?) and what is his experience? Just for completeness this should be added.

Author Response

Manuscript ID: animals-1226764

Dear Reviewer 3,

We appreciate the reviewers’ feedback and have addressed the reviewers’ comments. Our responses to the comments are listed below along with the locations of the changes made in the revised manuscript.

I have attached the revised manuscript entitled: “Changes of dorsal root ganglion volume in dogs with clinical signs of degenerative myelopathy detected by water-excitation magnetic resonance imaging”. Our revisions are highlighted in the manuscript.

Please contact me should you have any questions or suggestions regarding the revised manuscript.

We look forward to hearing from you.

Sincerely,

Eiji Naito

Joint Graduate School of Veterinary Sciences, Gifu University, Gifu, Japan

Comments from the editors and reviewers:

-Reviewer 3 Comment

One last comment: what is the role of the person performing the measurements ( E.N. line 140)? (Neurologist, radiologist etc?) and what is his experience? Just for completeness this should be added.

Our response:

Thank you for your comment. We agree with you and have incorporated this suggestion into the revised manuscript.

We added the following sentence:

E.N. was a practicing veterinarian who received training in veterinary radiology and neurology for three years.

This change has been made in the following location:

Materials and Methods, line 140-141

Other changes were described in the manuscript as follows:

Simple Summary, line 19 – change “dorsal root ganglion (DRG)” to “DRG”

Abstract, line 37 – change “r” to “rs

Materials and Methods, line 172 - change “r” to “rs

Results, line 228-234, 262 - change “r” to “rs